# Preventing the Collapse Behavior of Polyurethane Foams with the Addition of Cellulose Nanofiber

**DOI:** 10.3390/polym15061499

**Published:** 2023-03-17

**Authors:** Sanghyeon Ju, Ajeong Lee, Youngeun Shin, Hyekyeong Jang, Jin-Woo Yi, Youngseok Oh, Nam-Ju Jo, Teahoon Park

**Affiliations:** 1Composites Research Division, Korea Institute of Materials Science (KIMS), 797, Changwon-Daero, Seongsan-Gu, Changwon-si 51508, Republic of Korea; 2School of Chemical Engineering, Pusan National University, Busan 46241, Republic of Korea

**Keywords:** anti-collapse, polyurethane foam, cellulose nanofiber, hydrogen bonding, nucleation agent

## Abstract

Polyurethane foam manufacturing depends on its materials and processes. A polyol that contains primary alcohol is very reactive with isocyanate. Sometimes, this may cause unexpected problems. In this study, a semi-rigid polyurethane foam was fabricated; however, its collapse occurred. The cellulose nanofiber was fabricated to solve this problem, and a weight ratio of 0.25, 0.5, 1, and 3% (based on total parts per weight of polyols) of the nanofiber was added to the polyurethane foams. The effect of the cellulose nanofiber on the polyurethane foams’ rheological, chemical, morphological, thermal, and anti-collapse performances was analyzed. The rheological analysis showed that 3 wt% of the cellulose nanofiber was unsuitable because of the aggregation of the filler. It was observed that the addition of the cellulose nanofiber showed the improved hydrogen bonding of the urethane linkage, even if it was not chemically reacted with the isocyanate groups. Moreover, due to the nucleating effect of the cellulose nanofiber, the average cell area of the produced foams decreased according to the amount of the cellulose nanofiber present, and the average cell area especially was reduced about five times when it contained 1 wt% more of the cellulose nanofiber than the neat foam. Although the thermal stability declined slightly, the glass transition temperature shifted from 25.8 °C to 37.6, 38.2, and 40.1 °C by when the cellulose nanofiber increased. Furthermore, the shrinkage ratio after 14 days from the foaming (%_shrinkage_) of the polyurethane foams decreased 15.4 times for the 1 wt% cellulose nanofiber polyurethane composite.

## 1. Introduction

Since polyurethane (PU) foam was commercialized in the 1950s, it has emerged as a versatile material with diverse applications, such as cushioning, packaging, automotive interiors, furniture, bedding, and construction insulation [1]. PU foam is generally synthesized through the urethane formation reaction of poly-isocyanate and the hydroxyl group of a polyol. Additionally, various additives, such as surfactants, blowing agents, catalysts, and chain extenders are introduced to control the properties and cell morphology of PU foam [2,3]. The properties of PU foam vary depending on the raw materials that are used in its production. The mechanical properties, thermal properties, and density of PU foam are all influenced by factors such as the type and amount of polyol, isocyanate, and the other additives used [3,4,5,6].

Moreover, the free-rise method, which is commonly utilized in the industrial field for producing PU foams and was studied extensively in the literature [7,8], was implemented on a lab scale to produce PU foam insulation. For green chemistry, we tried to utilize the glycerol ethoxylate which consisted of highly reactive primary alcohols, in order to reduce the use of organic catalysts. However, the collapse of the synthesized PU foam was observed even post-curing. The PU foam was inflated when foaming but sank after a few days. For this reason, the gas that was generated by a chemical reaction grew into the pore; however, gas leakage occurred inside the cells [9]. This phenomenon is a critical issue in industrial product manufacturing. Previous studies have identified several causes of this issue, such as an insufficient temperature during the foaming, rapid gelation, and the absence of surfactant [3,10,11,12,13]. 

Cellulose is the most abundant natural polymer on our planet and has a variety of applications, such as composites, netting, coatings, and food packaging [14,15,16]. Cellulose can be derived from numerous sources, including woods, annual plants, algae, and bacteria [17,18,19]. It is composed of numerous glucose units that are linked by β-1,4-glycosidic bonds [16,20,21], and there are plenty of hydroxyl groups on the glucose units which can readily form hydrogen bonds with each other to hold the chain together [22]. Cellulose nanofiber (CNF) is an aggregation of 10–50 cellulose elements with a web-like network structure. It is obtained from cellulose through various methods, including high-pressure homogenization, grinding, electrospinning, and TEMPO-mediated oxidation pre-treatment. CNF has a range in diameter of 5 to 50 nm and a length of a few μm. [15,23].

Due to its reinforcement effect on the properties of composites, CNF has been studied widely as a filler. Iwatake et al., reported that the use of CNF improved the Young’s modulus and tensile strength of polylactic acid by 40% and 25%, respectively, without reducing the yield strain at a fiber content of 10 wt% [24]. Zhou et al. discovered that adding CNF of up to 0.5 wt% to PU foam increased the compressive strength and modulus of the foams without a significant effect on their density, porosity, or closed cell content [25]. Kamboj et al. found that adding nano cellulose to polyvinyl acetate adhesive significantly improved the joints’ elastic stiffness and shear strength [26]. Choo et al. identified that the tensile strength and thermal stability of polyvinyl alcohol-chitosan film were enhanced by increasing the loading levels of TEMPO-oxidized CNF to 1.5 wt% [27].

Not only was enhancing the effect of the mechanical properties of composites studied, but improving the dimensional stability of PU foams via the addition of cellulose derivative was also explored. Xiaojian et al. reported that the dimensional stability in freezing and heating conditions was increased by adding cellulose nanocrystals (CNC). Significantly, the volume shrinkage in the freezing conditions decreased from 5% to 1% when the CNC content was increased to 8 phr [28]. Additionally, Xiaojian et al. studied the addition of a pineapple leaf nanofiber to the compressive properties and dimensional stability of a biobased PU foam. Their research showed that the compressive strength and modulus increased by 35% and 23% with 2% nanofiber, and the dimensional stability significantly improved in the biobased PU foam [29].

In this study, the aim was to investigate the effect of incorporating CNF as a filler material to prevent the synthesized PU foam’s collapse behavior. The foams were prepared using the free-rise method and were post-cured before being cut into a specific size. It was identified that there is an actual effect in preventing the collapse behavior of the foams. This can be attributed to the change in the chemical linkage and eco-friendly nucleating agents’ effects caused by the incorporation of CNF. The effectiveness of adding CNF as a filler was evaluated through several experiments, including rheology, FTIR, thermal analysis, morphology, and a collapse behavior assessment.

## 2. Materials and Methods

### 2.1. Materials

Glycerol ethoxylate (average M_n_~1000) with a hydroxyl value of 160–177 mg KOH mg^−1^ was used as a polyol. The fibrous cellulose from cotton linter was used as the raw material for the fabrication of the CNF. Both the polyol and cellulose raw material were purchased from Sigma-Aldrich (St. Louis, MO, USA). Methylene diphenyl diisocyanate (Lupranat M 20 S), with an NCO content of 31.5%, a viscosity of 210 mPa·s at 25 °C, and a functionality of 2.7 for polymeric MDI (pMDI), was obtained from BASF SE (Ludwigshafen, Germany). Silicone surfactant (OFX-193S) was kindly supported by Dow Chemical (Midland, MI, USA). Dibutyltin dilaurate (DBTDL, 95%) catalyst was purchased from TCI (Tokyo, Japan). Triethanolamine (99%) for the chain extender was obtained from SAMCHUN Chemicals (Seoul, Republic of Korea). Distilled water was used as a blowing agent.

### 2.2. Preparation of CNF

The cellulose fibrous was dispersed in water at room temperature with a spatula, with a weight ratio of 1:100 (cellulose:water). When it became homogeneous, the suspension was treated with a high-pressure homogenizer (NH500, ILSHIN Autoclave, Daejeon, Republic of Korea) in 1000 bars for 10 passes to get the CNF. Finally, the CNF suspension was dried via freeze drying (FDCF-12012, Operon, Seoul, Republic of Korea) for 7 days at −130 °C in a vacuum.

### 2.3. Preparation of Foams

There is an overview of the foam preparation in Figure 1. The CNF was pulverized with a rotor mill (PULVERISETTE 14, Fritsch, Idar-Oberstein, Germany) at 12,000 rpm, with a sieve of a 120 μm mesh aperture. After that, the CNF powder was dried for 4 h in a 60 °C oven to evaporate the water. The CNF was added into the polyol, and then mixed at 1400 rpm for 5 min with a planetary mixer (HGT 400-DIV, HANIL Global Technology, Seoul, Republic of Korea). The rheological properties were measured in this step. When the mixture became homogeneous, the water, chain extender, surfactant, and catalyst were added to the mixture, mixed at 1400 rpm for 3 min, and then defoamed at 1400 rpm for 3 min in a vacuum. The neat foam was also prepared in this step. After that, MDI was added and vigorously stirred for 10 s with a disperser (IKA T50 Digital ULTRA-TURRAX^®^, IKA works, Staufen im Breisgau, Germany). Then, the mixture was poured into an open paper mold (11 × 11 × 2.5 cm) to produce free-rise foam. The foaming began after 10 to 15 s, and the entire process was finished in 90 s. The PU foams were cured at room temperature for 24 h [30] and elaborated into specific sizes before the characterization. The formulations are shown in Table 1. The hydroxyl group of the CNF was also counted in the -NCO/-OH index value. The PU foams with 0, 0.25, 0.5, 1, and 3% of the CNF (based on the total Pbw of the polyols) were named N, C0.25, C0.5, C1, and C3, respectively.

### 2.4. Rheological Properties

The rheological properties of the polyol and CNF in the polyol mixture were characterized using a modular compact rheometer (Anton Paar, MCR 302, Graz, Austria) at 20 °C in a shear rate from 0.1 s^−1^ to 100 s^−1^, with a diameter of a 25 mm parallel plate and a gap of 0.5 mm.

### 2.5. Attenuated Total Reflection-Fourier Transform Infrared (ATR-FTIR) Spectroscopy

The ATR-FTIR spectra were obtained using an FTIR spectrometer (Thermo Nicolet, Nicolet iS10 FT-IR Spectrometer, Madison, WI, USA) in ATR mode. The spectra were collected from 4000 to 600 cm^−1^, with 32 scans at a resolution set to 4 cm^−1^. The foams were pulverized with a rotor mill and measured to be about 0.1 g of the samples in powder form. The spectra were normalized with the phenyl group of 1595 cm^−1^ [31].

### 2.6. Morphology Characterization

The field-emission scanning electron microscope (FE-SEM) image of the CNF was obtained using FE-SEM (Jeol, JSM-7900F, Seoul, Republic of Korea). The FE-SEM examination at 20,000 magnifications was operated for the platinum sputter-coated CNF powder at a 3.0 kV accelerating voltage. The scanning electron microscope (SEM) image of the PU foams was obtained using a tabletop SEM (SEC, SNE-4500M Plus, Seoul, Republic of Korea). This observation was conducted on the foam one day after the foaming and before the collapse behavior occurred. The cross-section, parallel to the foaming direction of the foam, was sputter-coated with platinum. The SEM examination at 74 magnifications was operated at a 5.0 kV accelerating voltage. The cell size was calculated with Image J freeware (Appendix A).

### 2.7. Differential Scanning Calorimetry (DSC) and Thermogravimetric Analysis (TGA)

Differential scanning calorimetry was conducted using the DSC (TA Instruments, Q2000, Newcastle, DE, USA). The samples were heated from −40 to 150 °C at a heating rate of 5 °C/min under air conditions. Approximately 4–5 mg of the samples were prepared for each measurement. The thermogravimetric analyses (TGA) of the foams were conducted using a thermogravimetric analyzer (TA Instruments, TGA 55, Newcastle, DE, USA). The temperature ranged from 30 to 700 °C with a heating rate of 10 °C/min under the air atmosphere.

### 2.8. Collapse Behavior Characterization

The foams were cut into specific sizes and their weights were measured on the 1st day and 14th day. The density of the samples was also obtained by using an analytical balance (Mettler Toledo, PG603-S, Columbus, OH, USA) for each day. The volumes of the PU foams on the 1st day (*V*_0_) and 14th day (*V*_n_) were calculated using the measured weight and density of the PU foams. The ratio of the shrinkage (%_shrinkage_) was calculated using Equation (1):(1)$${\%}_{\mathrm{shrinkage}}=\frac{V_{0}-V_{n}}{V_{0}}\times 100\%$$

## 3. Results and Discussions

### 3.1. FE-SEM Examination of CNF

Due to the pulverizing and drying process, it is necessary to confirm the structure of the treated CNF. The lyophilized CNF powder and its FE-SEM image are shown in Figure 2a,b. Through the rotor milling process, the freeze-dried CNF was pulverized and could be obtained as a white and delicate CNF powder. In addition, the FE-SEM image shows that the fibrous structures and networks of the CNFs are well maintained after being powdered.

### 3.2. Influence of the Amount of CNF Added on Rheological Properties

The rheological properties were used to analyze the appropriate CNF concentration in the polyol. This measurement was conducted on a polyol and a mixture of the CNF and polyol in the foaming process. Figure 3 shows the viscosity of the polyol and the polyols containing CNF. The viscosity of the polyol was increased with the CNF loading at the same shear rates, especially in C3. For example, the dispersion of 3 wt% of the CNF in the polyol produced a 16 times higher shear viscosity than that of the neat polyol at a low shear rate (0.1 s^−1^), while the 0.25 wt% of the CNF added into the polyol showed almost no difference to the neat polyol. In the case of C0.5 and C1, their viscosity was increased 1.3 and 1.6 times, respectively. This noteworthy increase in viscosity at the low shear rate in C3 was because the fibers were well dispersed in the matrix and had formed networks. Only C3 generated these networks because the increase of the CNF concentration led to binding and the creation of a structural percolation network [32,33].

Additionally, the difference in the shear thinning behavior according to the CNF concentration indicated this. In the case of the polyols with up to 1 wt% of the CNF, there was no significant decrease in the viscosity with an increase in the shear rate, whereas C3 showed significant shear thinning behavior (about 7 Pa·s to 1 Pa·s). This is because the shear force caused the breakdown of the entangled CNF network into scattered nanofibers domains, leading to a decrease in viscosity [34]. However, this network partially recovered and caused spatial defects, consequently generating the collapse of the foam during curing, as shown in Appendix A. Therefore, further experiments were carried out with the samples C0.25, C0.5, and C1, in which the collapse (caused by excess CNF) did not occur.

### 3.3. ATR-FTIR Analysis

The attenuated total reflection-Fourier transform infrared spectroscopy was conducted to analyze the changes in the chemical bonds of the PU foams that were caused by adding the CNF. Figure 4a shows the ATR-FTIR spectra of the PU foams with varied amounts of 0, 0.25, 0.5, and 1 wt% of the CNF. The large absorption peak of the NH stretching and bending vibration at 3320 cm^−1^ and 1530 cm^−1^ confirms the urethane linkage [35]. Furthermore, this was confirmed also by the C-H stretching peak at 3000–2800 cm^−1^, the isocyanate absorbance at 2275 cm^−1^ [36], the carbon dioxide vibration band at 2360 cm^−1^ [37], and the phenyl group of pMDI centered at 1595 cm^−1^ [31]. Because the foaming gas was carbon dioxide, there is a slight difference at 2360 cm^−1^ among the samples, regardless of the chemical bond.

In Figure 4b,c, the free carbonyl stretching band at 1724 cm^−1^ [38] and the C-O-C peak at 1069 cm^−1^ [39] of the urethane group are decreased with the increasing CNF content. This is because, as mentioned above, the hydroxyl group of the CNF was counted in the -NCO/-OH index. Therefore, as the amount of the CNF increased, the amount of the polyol decreased. However, since the CNF did not participate in the urethane synthesis, the peak of the urethane group was reduced. This tendency can verify that the isocyanate peak remains as the CNF is added [40], as shown in Figure 4d.

In contrast with the results of several studies [41,42], the CNF did not react with the isocyanate in this research. This is because the polyol used in this paper is a primary alcohol that is highly reactive, rather than a secondary alcohol. Polyols containing primary alcohol are usually three times more reactive with isocyanate than secondary alcohol [43]. Therefore, a decreased preference for the hydroxyl group of the CNF meant that it could not react with the isocyanate to form a urethane structure. Additionally, the reactivity of the solid was reduced when in the liquid phase [44]; thus, the hydroxyl group of the CNF did not react with the -NCO of pMDI. Despite all this, the H-bonded urethane carbonyl stretching band at 1707 cm^−1^ [45] is increased along the CNF content in Figure 4b. This means that the cellulose surface hydroxyl groups provided the site to make an H-bond with the carbonyl groups of the urethane bond [46,47].

Consequently, these increased H-bond sites prevented the collapse behavior of the PU foams. Leng et al. also reported similar results [3]. Although they were not notified about the reactivity of the polyol, the FTIR spectra showed that the CNF did not introduce any new functional groups into the polyurethane using a polyol that consisted of primary alcohol.

### 3.4. Morphology of Foams

In addition to chemical bonding, cell morphology also played an essential factor in the collapse behavior of foam. The foam cell characteristics of the N, C0.25, C0.5, and C1 on day one were studied using SEM. Figure 5 shows the cell morphology of the different CNF loading. It was found that the PU foam had a smaller cell size depending on the increases to the amount of CNF. Due to the anisotropic structure of the cell, the diameter is not a suitable measurement for determining its size. Instead, the average cell area was calculated with Image J freeware, as shown in Figure 6. Samples N, C0.25, C0.5, and C1, treated with increasing concentrations of the CNF, exhibited decreasing cell areas in the order of N (>100,000 μm^2^), C0.25 (~37,000 μm^2^), C0.5 (~30,000 μm^2^), and C1 (~21,000 μm^2^), respectively. Additionally, the addition of the CNF resulted in a more uniform foam structure.

This observed tendency with increases to the concentrations of the CNF suggests that the CNF had a significant effect on the foam as a nucleating agent. The added CNF particle acted as a nucleating agent and provided a site for gas bubble formation instead of the coalescence of gas [48,49,50], thereby reducing the size of the cell and promoting a more uniform foam structure. Accordingly, the average cell area was about 5 times smaller for C1 than N. Hence, like the increased H-bonded sites, the incorporation of CNF as a nucleating agent was found to significantly reduce the coalescence of gas and facilitate the formation of smaller and homogeneous cells, thus preventing the collapse behavior of the PU foam.

### 3.5. Thermal Properties of PU Foams

The effect of applying CNF to PU foams on their thermal properties was investigated using DSC and TGA. Figure 7 shows the glass transition temperature (T_g_) of the hard segment in the PU foams, and it was found that the T_g_ of sample N was around 25.8 °C, while the T_g_ of samples C0.25, C0.5, and C1 were around 37.6 °C, 38.2 °C, and 40.1 °C, respectively. These results suggest that the addition of the CNF led to an increase in the T_g_, with a tendency to increase as the amount of CNF increased. The T_g_ was affected by the morphological structure of the foams and the H-bonding that formed among the urethane groups [46]. Thus, the incorporated CNF induced a finer cell structure and additional H-bonding sites; consequently, the T_g_ of the PU foams was increased. This result is in good agreement with the FTIR and morphology analyses.

The TGA and DTG curves of the sample are in Figure 8. As shown in Figure 8a, the TGA data presented the typical curves of the PU foam with two distinct steps. At first, there was a little weight loss between 85–100 °C that was related to humidity loss. After that, the urethane linkages and CNF were degraded around 300 °C (first stage), and the isocyanate component of the PU foams was broken down around 530 °C (second stage) [51,52]. Although the CNF was included, the TGA curves were not very different from those of the neat foam, since the amount of CNF was inadequate to change the overall shapes of the curves; however, slight alterations in the DTG curves are shown in Figure 8b. There was a tendency to increase the peak intensity and shift to a low temperature with the increasing CNF around 320 °C. These results are due to the decomposition of cellulose [53]. Additionally, the residue of the CNF acted as a defect and decreased the thermal stability, so that the peaks around 530 °C were shifted slightly to a lower temperature [51].

### 3.6. Dimensional Stability

Each sample was prepared twice for non-destructive (photographs in Figure 9 and weighing) and destructive experiments (density measurements in Table 2), and all the PU foams were kept in the same condition. The density of the foams was measured with a density meter. Since the foam was prepared closed-cell, the water absorption could be negligible [12]. The foam volumes were calculated using the density and weight of the samples on days 1 and 14, and a photo shoot was also carried out each day. This method offers further accuracy for measuring the volume of un-regular shapes.

In Figure 9, the results show a noteworthy change in the collapse behavior of the PU foams with the application of CNF. Compared with the neat foam and the CNF/PU composite foams, the N completely collapsed in two weeks, whereas the C0.25, C0.5, and C1 almost maintained their original shapes, even after two weeks. In order to identify the detailed dimensional stability of each sample, the collapse behavior features on day 1 and day 14 are summarized in Table 2. The initial density of the PU foam was 56.5 kg/cm^3^ and those of the CNF/PU composite foams were 50.7, 50.5, and 48.5 kg/cm^3^, according to the increasing amount of the CNF. This is in line with the general phenomenon of urethane, in which the density decreases as the pore size decreases.

Additionally, the %_shrinkage_ of N was recorded at 32.48%, while at 2.80, 2.74, and 2.11% for C0.25, C0.5, and C1, respectively. As mentioned above, the decrease in the %_shrinkage_ of the CNF-incorporated PU foams was due to the CNF providing hydrogen bonding with the carbamate groups in urethane linkage and acting as a nucleating agent to make a uniform and smaller cell structure. In the case of the PU foam, the higher the density, the higher its shape retention stability. However, it was confirmed that adding nanofillers could bring about the shape stability effect independently of the density. As a result, the PU foams reinforced with the CNF had significantly improved anti-collapse performances. Conversely, sample N, which was not enhanced with the CNF, collapsed from the cell structure, as shown in Appendix A.

## 4. Conclusions

In this study, a CNF suspension was fabricated by a high-pressure homogenizer in order to maintain the hydroxyl groups on the CNF main chains, and it was lyophilized to obtain CNF powder. Then, it was applied to semi-rigid PU foam to prevent the collapse behavior of the foam. The hydroxyl groups on the surface of the CNF could make H-bonding with the carbamate groups in urethane linkage, even if the direct reaction with the isocyanate group did not occur. The reactivity of the polyol and CNF with the isocyanates played a key role in the urethane reaction. In addition, the CNF acted as a nucleating agent, making the cell size small and homogenous. The rheology properties were measured to verify the dispersity and optimized concentration of the CNF. For these reasons, the T_g_ of the CNF-added PU foams shifted to a high temperature, and the %_shrinkage_ of the foam was reduced by 15.4 times. This study suggests a novel method of preventing the collapse of synthesized PU foam by adding a small amount of CNF (only 0.4 wt% of the total weight of a CNF/PU composite).

## Figures and Tables

**Figure 1 polymers-15-01499-f001:**
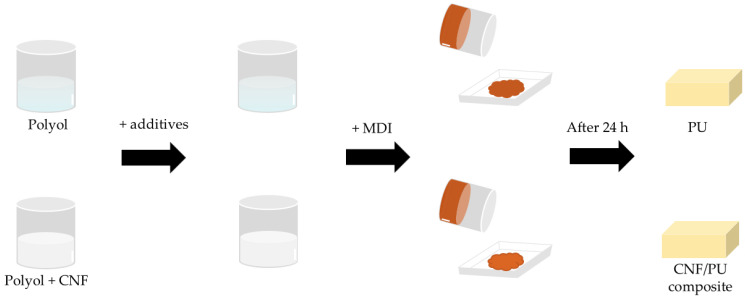
Simple schematic image of foam preparation.

**Figure 2 polymers-15-01499-f002:**
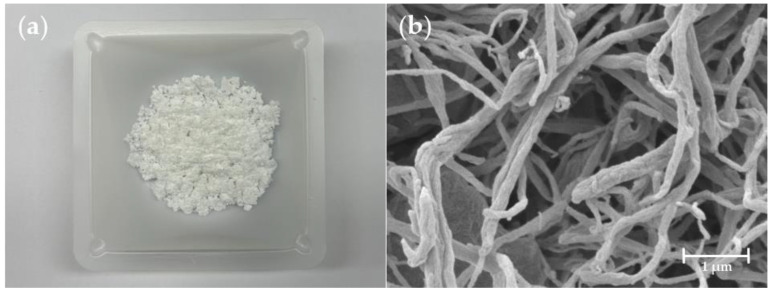
(**a**) CNF powder, and (**b**) FE-SEM image of CNF powder.

**Figure 3 polymers-15-01499-f003:**
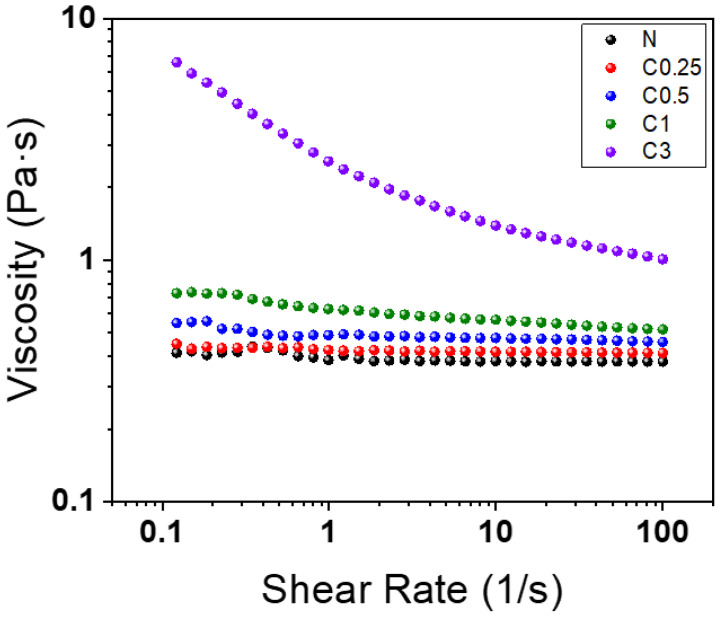
Rheometer curves of polyol and polyols after mixing with CNF.

**Figure 4 polymers-15-01499-f004:**
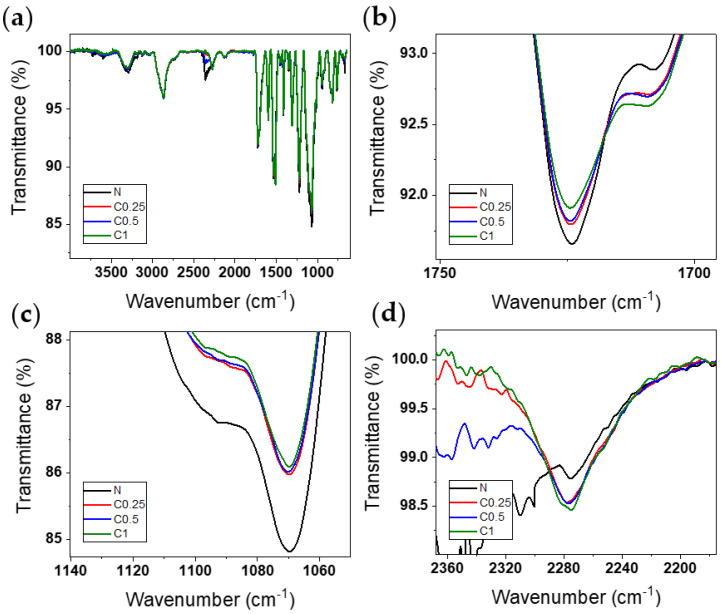
FTIR spectra of PU foams, (**a**) expanded view of the spectra, (**b**) carbonyl peaks of urethane group at 1724 cm^−1^ and 1707 cm^−1^, (**c**) C-O-C peak of urethane group at 1069 cm^−1^, and (**d**) isocyanate peak of pMDI at 2270 cm^−1^.

**Figure 5 polymers-15-01499-f005:**
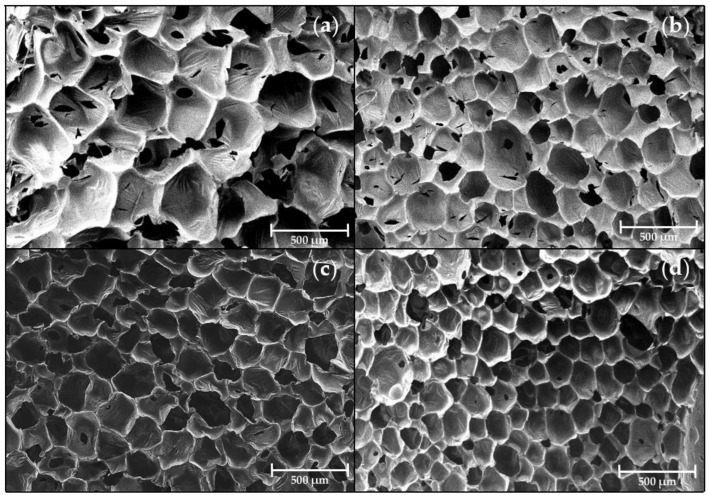
SEM image of prepared foams, (**a**) N, (**b**) C0.25, (**c**) C0.5, and (**d**) C1.

**Figure 6 polymers-15-01499-f006:**
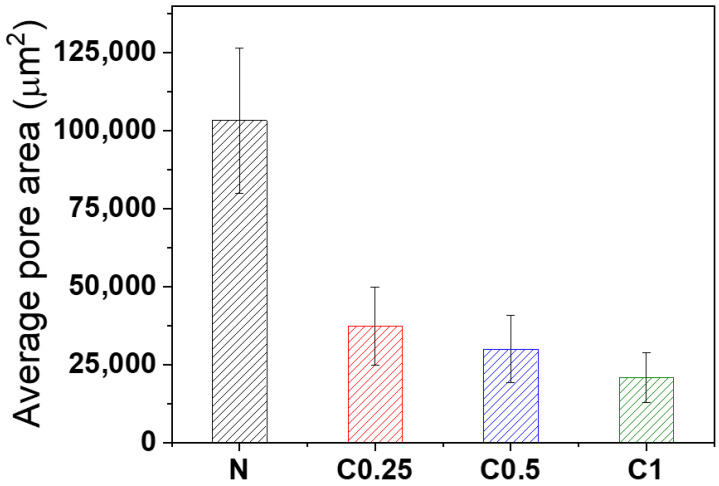
The average cell size of foam with different CNF contents compared with neat foam.

**Figure 7 polymers-15-01499-f007:**
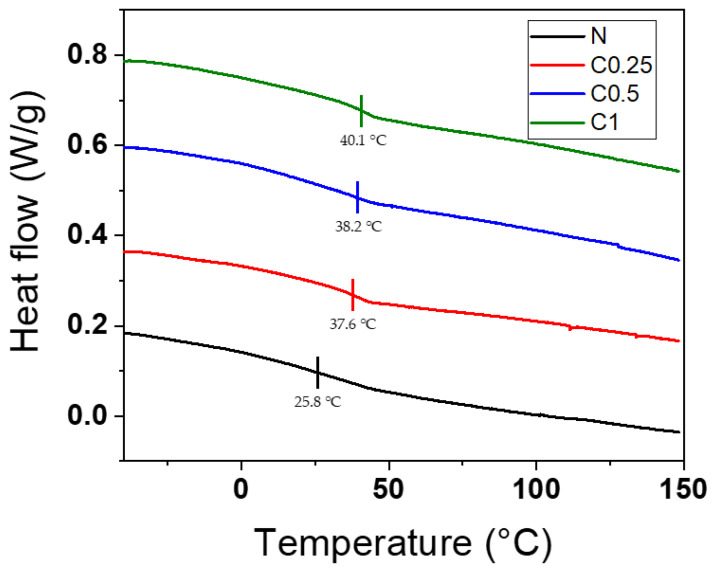
DSC curves of N, C0.25, C0.5, and C1.

**Figure 8 polymers-15-01499-f008:**
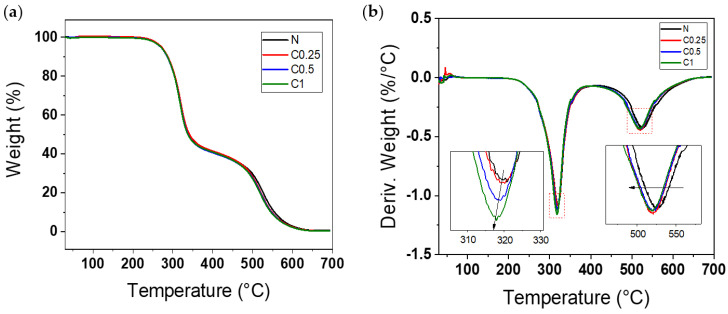
(**a**) TGA, and (**b**) DTG curves of PU foams in air atmosphere.

**Figure 9 polymers-15-01499-f009:**
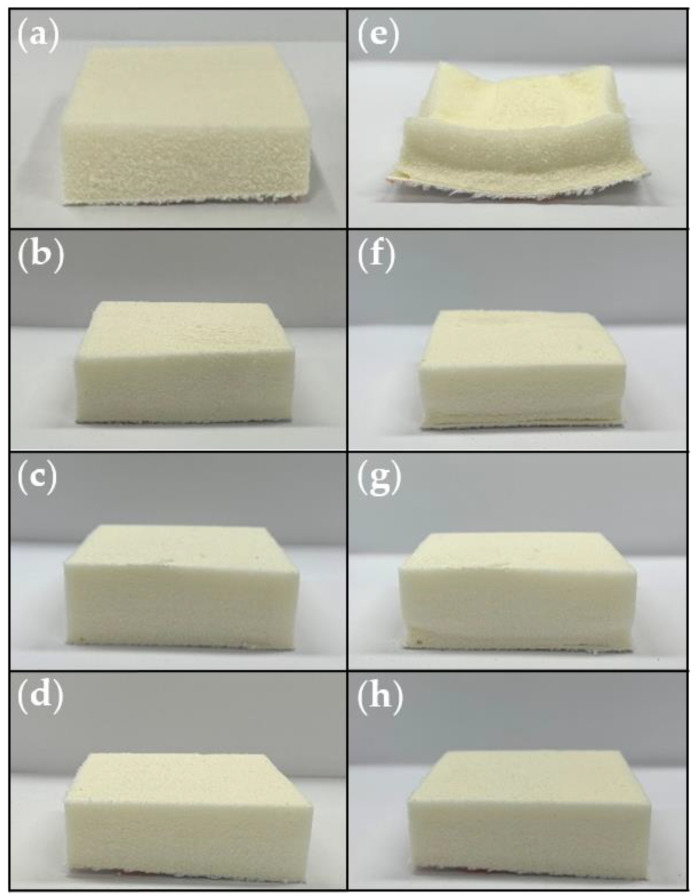
The picture of PU foams with samples (**a**) N, (**b**) C0.25, (**c**) C0.5, (**d**) C1 at day 1, and (**e**) N, (**f**) C0.25, (**g**) C0.5, and (**h**) C1 at day 14.

**Table 1 polymers-15-01499-t001:** The formulation of the PU foams.

Components	Parts by Weight (Pbw)	Role
Glycerol ethoxylate	100	Polyol
Lupranat M 20 S	126 (for N)	Poly-isocyanate
OFX-193S	4	Surfactant
DBTDL	0.5	Catalyst
Triethanolamine	3	Chain extender
Water	3	Blowing agent
CNF	0, 0.25, 0.5, 1, 3	Filler
Index (-NCO/-OH)	1.1	

**Table 2 polymers-15-01499-t002:** The collapse behavior assessment of PU foams.

Foams	%_shrinkage_	Density (kg/cm^3^)	
		Day 1	Day 14
N	32.48	56.5 ± 2.1	82.7 ± 5.4
C0.25	2.80	50.7 ± 0.5	51.5 ± 0.5
C0.5	2.74	50.5 ± 2.1	51.3 ± 0.9
C1	2.11	48.5 ± 0.5	49.0 ± 1.0

## Data Availability

Not applicable.

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
