# Peer review of "Preventing the Collapse Behavior of Polyurethane Foams with the Addition of Cellulose Nanofiber"

_polymers, 2023, doi:10.3390/polym15061499_

Round 1

Reviewer 1 Report

There are several comments.

1) Line 24. The designation "%shrinkage" can be used after decryption. In the Abstract section it is better to write this term in full.

2) Section Introduction. Since the research is connected, first of all, with the prevention of PU collapse, in the Introduction we would like a more detailed description of references on this topic. What is the collapse? How can it be prevented? In particular, have cellulose derivatives been used in PU composites before? With what result? The paragraph detailing the properties of a CNF looks optional.

3) Line 99. Unnecessary brackets.

4) Table 1. pMDI is not a role, but an abbreviated name. The role in this case is poly-isocyanate.

5) Figure 3a. What is the intense peak between 2500 and 2250 cm-1? It is observed only for sample N? If yes, why? If it is observed for all samples, then you need to insert an inset with an enlarged image of this area, as is done in Fig. 7b.

6) Lines 193, 235. Incorrect use of capital letters.

7) Line 237. Error in reference brackets.

8) Why, starting from the rheology section, there are no results for sample C3 with the maximum amount of CNF? Was it impossible to make an SEM due to the collapse? But the collapse would not interfere with the analysis by FTIR and DSC-TGA methods.

9) Figure 9. This figure should either be removed in the supplementary file or added with an SEM photograph after 14 days for one of the PU-CNF samples.

Author Response

Reply to the comments from Reviewers

The authors appreciate the time and effort that the reviewers dedicated to providing feedback and are thankful for the insightful comments and valuable suggestions provided. Each point has been revised carefully and the manuscript is revised accordingly. All changes are highlighted in red color in the manuscript and the supporting information. 

Reviewer 2 Report

The paper by Ju S. et al. examines the use of nanocellulose to improve the stability of polyurethane foam. The authors obtain nanocellulose, add it to the initial reagents, synthesize the foam, and study its morphological features, thermophysical properties, and density as a function of nanocellulose concentration. As a result, the authors show that 0.4% nanocellulose allows for a significant improvement in foam stability. It is a good work that needs a few corrections before publication.

Specific comments are as follows.

Line 79: “Commercial cellulose fibrous was used as filler material.” Please specify in what form the cellulose was (powder, aqueous dispersion, or something else).

Line 88: “Commercial cellulose fibrous was used as filler material.” This sentence has no subject and no predicate. In addition, please indicate the shear rate that was used in treating the cellulose, the temperature, and the duration of the treatment.

Line 96: "After that, CNF powder was dried 96 for 4 hours in a 60 °C oven to evaporate water." It is necessary to provide an SEM image of the obtained CNF powder.

Line 105: "to produce free-rise foam." Please indicate how quickly the foam formation was.

Line 139: Please write the meaning of ?0 and ?n.

Line 148-156: “This increased viscosity along the added amount of CNF because the fibers are well dispersed in the matrix and have formed networks [29-31]. It is because random orientation is maintained when the cellulose concentration is low, but when it is high, CNF is oriented according to the shear force and has a non-Newtonian behavior [32,33]. However, this orientation leads to a tendency to aggregation of CNF [34]”. This is a wrong interpretation. At a CNF content of 0.25-1%, the cellulose fibrils do not form a network (their concentration is below the percolation threshold), and therefore the viscosity of the dispersion does not depend on the shear rate. When 3% CNF is added, the cellulose fibrils agglomerate to form a spatial network giving the dispersion a yield stress behavior with the yield strength. The transition from low shear rates to high shear rates increases the shear stress, destroying the structural network of cellulose fibrils at first and then the destruction of cellulose agglomerates. In other words, the higher the shear rate, the more agglomerates of cellulose (flocs) are destroyed and the lower the viscosity.  The viscosity decreases because the agglomerates break down, not because of the orientation of the cellulose fibrils. There is no orientation of the cellulose fibrils under the action of shear force. This is a myth, as otherwise shear-thinning behavior would be in less concentrated samples without a network, but in which the orientation of the cellulose fibrils could be (for the orientation of particles under the action of the shear field, the dispersion medium must have certain viscoelastic characteristics, whereas the orientation of small particles in a low-viscosity fluid is impossible in principle since the high shear rates that could cause orientation will instead cause turbulence). This paragraph should be rewritten. More information can be found in recent articles on this topic (10.1007/s10570-019-02908-w, 10.1021/acs.energyfuels.0c02797, 10.1007/s10570-021-03745-6, 10.1016/j.triboint.2022.108080).

Line 156: "However, this orientation leads to a tendency to aggregation of CNF [34] and results in the collapse of foam during curing  [35] as shown in Figure S2." As noted above, there can be no orientation of fibrils. Unlike other systems, a system with 3% cellulose has a structural network of cellulose fibrils. This network probably caused the collapse of foam during curing. Perhaps the network causes spatial defects or makes the system brittle.

Line 160: "Rheological properties". There are many rheological properties, whereas only the dependences of viscosity on shear rate are given here. This needs to be clarified.

Author Response

(The authors gave the same response as above.)

Round 2

Reviewer 2 Report

The authors have made all the necessary corrections to the text of the manuscript, and it can now be published. However, one change must be made:

Line 336: “CNF solution” -> “CNF dispersion”. Cellulose nanofibers are dispersed in water rather than dissolved in it.